# Lipoprotein Profile in Populations from Regions of the Russian Federation: ESSE-RF Study

**DOI:** 10.3390/ijerph19020931

**Published:** 2022-01-14

**Authors:** Victoria A. Metelskaya, Svetlana A. Shalnova, Elena B. Yarovaya, Vladimir A. Kutsenko, Sergey A. Boytsov, Eugeny V. Shlyakhto, Oxana M. Drapkina

**Affiliations:** 1National Medical Research Center for Therapy and Preventive Medicine, 101990 Moscow, Russia; sshalnova@gnicpm.ru (S.A.S.); yarovaya@mech.math.msu.su (E.B.Y.); vlakutsenko@ya.ru (V.A.K.); odrapkina@gnicpm.ru (O.M.D.); 2Department of Probability Theory Faculty of Mechanics and Mathematics Lomonosov Moscow State University, 119234 Moscow, Russia; 3National Medical Research Center of Cardiology, 121552 Moscow, Russia; prof.boytsov@gmail.com; 4Almazov Federal Medical Research Center, 197341 St. Petersburg, Russia; e.shlyakhto@almazovcentre.ru

**Keywords:** dyslipidemia, ESSE-RF, lipoprotein profile, population-based study, prevalence, socioeconomic status

## Abstract

This study aimed to describe the dyslipidemia prevalence and pattern among adult populations from different regions (*n* = 13) of the Russian Federation (RF). Randomly selected samples (*n* = 22,258, aged 25–64) were studied according to the ESSE-RF protocol. Lipoprotein parameters were estimated by routine methods. Statistical analyses were performed using R software (v.3.5.1). The overall dyslipidemia prevalence was 76.1% (76.9/75.3% for men/women). In women, total cholesterol (TC) and low-density lipoprotein (LDL)-C levels gradually increased with age (from 4.72 to 5.93 and from 2.76 to 3.79 mmol/L, respectively); in men, they reached a maximum by 45–54 (5.55 and 3.55 mmol/L, respectively) and then decreased. No differences in high-density lipoprotein (HDL)-C in men of different ages were found, but slight decreases in HDL-C and apo AI were observed in women by 55–64 years. No pronounced associations between education and lipid levels in men were observed; higher-educated women showed significantly better lipoprotein profiles. Similar associations between lipids and income level were detected. Women from rural areas had higher TC and triglycerides than urban residents. Regardless of sex, rural residents had higher HDL-C and apo AI, and reduced apo B/apo AI. Conclusion: Information on the peculiarities of dyslipidemia prevalence and lipoprotein profile depending on sex, age, residential place, and socioeconomic status is useful for assessing the global ASCVD risk, and for risk modeling based on national data.

## 1. Introduction

Atherosclerosis-related cardiovascular diseases (ASCVDs) are the leading cause of death: about 17.9 million people died from these diseases in 2019, representing 32% of all global deaths [1]. According to epidemiological, clinical, and experimental studies, dyslipidemia (serum lipid transport abnormalities) is regarded as one of the most powerful modifiable ASCVD risk factors, and people having this disorder are at increased risk of developing these diseases as compared to normolipidemics [2].

Epidemiological studies are an important scientific tool for assessing the prevalence of ASCVDs and their risk factors, as well as predicting their outcomes. The ESSE-RF (Epidemiology of Cardiovascular Diseases and their Risk Factors in Some Regions of the Russian Federation) study is the first population study conducted in Russia during the last two decades, which is considered as a continuation of preventive activity, in order to obtain unbiased information about the epidemiological characteristics of the population of some regions of Russia [3,4,5]. Based on the ESSE-RF study, the prevalence and geographic distribution of several ASCVD risk factors, including overweight and obesity, arterial hypertension, smoking, and some trends in dyslipidemia in Russian adults aged 25–64 years had already been evaluated and discussed [3,6,7,8]. At the same time, a detailed analysis of various dyslipidemias, depending not only on sex and age, but also on socioeconomic status (levels of education and income, place of residence), has not been published yet.

Previously reported results of the analyses of trends in serum lipid levels and differences in dyslipidemia characteristics were limited to several countries, without a consistent and comparable global analysis. Along with the universal nature of the influence of the main risk factors on ASCVD development, it is obvious that the impact of each of these risk factors varies significantly depending on the country and region. Indeed, there are data in the literature describing the specific features of the prevalence and pattern of dyslipidemia in various countries [9,10]. As a whole, significant changes in total cholesterol (TC) and potentially atherogenic lipid fractions have been found, demonstrating a decline in TC in high-income Western countries and its rise in low- and middle-income countries in some regions of Asia and Latin America [11].

It should be noted that, in Russia, there is a shortage of data for predicting cardio-vascular morbidity, and the use of foreign models is limited by the complexity of extrapolating their results to the Russian population. The knowledge about the prevalence of different types of dyslipidemia, as well as data on mean values of particular lipoprotein parameters and their peculiarities associated with socioeconomic conditions, will be useful both for cardiovascular risk modeling based on our own epidemiologic data and for comparison of the epidemiological situation in Russia with other countries.

This study aimed to describe the dyslipidemia prevalence and pattern among adult populations from different regions of the Russian Federation (RF), studied according to the ESSE-RF protocol.

## 2. Materials and Methods

A multicenter cross-sectional population-based ESSE-RF study (Clinicaltrials.gov NCT04306822) was conducted in representative samples from the free-living population aged 25–64 years (*n* = 22,258, 37.4% men) from 13 regions of the Russian Federation, differing in climate and geographic, economic, and ethnic parameters: the Republic of North Ossetia (Alania) (North Caucasus), Volgograd (South), St. Petersburg and Vologda (North West), Voronezh and Ivanovo (Central), Orenburg and Samara (Volga Region), Tyumen (Ural), Kemerovo and Tomsk (West Siberia), Krasnoyarsk (East Siberia), and Vladivostok (Far East), with geographic region in brackets.

The design of the ESSE-RF study was reported in an earlier study [3]. Briefly, the study used a multistage clustered sample design based on district outpatient departments (polyclinics) that were selected randomly as primary sampling units. This is a survey of the general population or free-living population, as the majority of the population are covered by obligatory health insurance. These primary sampling units covered neighborhoods with 30,000–80,000 adult residents. One polyclinic in each of the 13 regions was located in a rural setting; all others were located in urban areas. In each polyclinic, we randomly selected client lists from 5 physicians working there, to serve as secondary sampling units. Each secondary sampling unit had approximately 2000 adults aged 25–64 years. We then randomly selected 100 households from each secondary sampling unit. The total sample identified was approximately 2000 subjects in each of the 13 regions. Participants were recruited through letters and through phone, asking them to visit the clinic in the morning in a fasting state.

The study was conducted according to the guidelines of the Declaration of Helsinki and approved by the ethics committees from 3 Federal Centers where laboratory tests were performed: National Research Center for Preventive Medicine, the Russian Cardiology Research-and-Production Complex, and the Almazov Federal North-West Medical Research Centre. All participants signed an informed consent form before any measurements were collected. The overall response rate was about 80%.

The participants of the study were interviewed using a standard questionnaire developed on the basis of adapted international methods. The questionnaire was built according to a modular type and contained information about socio-demographic characteristics, behavioral habits, histories of the diseases, and economic living conditions. Education was determined as primary, secondary, or higher; residential location was determined as urban or rural. The income was assessed indirectly based on the answers of the study participants to questions concerning the share of income spent on food, as well as their opinion about the family budget and prosperity as compared to other families. To each question, 5 possible answers were offered, which were evaluated in points from 1 (“the poorest”) to 5 (“the richest”). Based on the sum of points, the income level was ranked into 3 categories (corresponding to the terciles of the distribution by points): low, middle, and high [8].

Blood was drawn at the regional polyclinics after patients’ overnight fasting using the standard technique. Serum was obtained by low-speed centrifugation (1000× *g*, 20 min, at temperature +4 °C). Serum samples after aliquoting in 500–1000 μL Eppendorf tubes were frozen at −25 °C and, within two weeks, were shipped on dry ice from the region to the corresponding Federal Center, where they were stored at −70 °C until analyses. Analyses were performed at the central standardized laboratories using an Abbott Architect c8000 autoanalyzer with Abbott Diagnostic kits (USA). TC, low-density lipoprotein (LDL)-C, HDL-C, and triglycerides (TG) were measured by enzymatic methods. Additionally, in some regions, we were able to measure apolipoproteins (apo) AI and B (*n* = 11,851), and lipoprotein (a) [Lp(a)] (*n* = 10,430). Apo AI and apo B concentrations were measured by an immunoturbidimetric assay; Lp(a) level was determined with a latex particle-enhanced turbidimetric immunoassay, which does not depend on the apo(a) isoform size.

Hypercholesterolemia was diagnosed at the level of TC ≥ 5.0 mmol/L and/or LDL-C ≥ 3.0 mmol/L; hypertriglyceridemia was determined by the TG level ≥1.7 mmol/L; hypoalphacholesterolemia was considered the level of HDL-C ≤ 1.0 mmol/L in men and ≤1.2 mmol/L in women. Apo B/apo AI ratio ≥ 1.0 and Lp(a) level ≥ 30 mg/dL were regarded as elevated ones.

Statistical analyses were performed using R software (version 3.5.1). Continuous variables are presented as mean (M) and standard deviation (SD) if Pearson’s nonparametric skewness coefficient was <0.2. Skewed parameters are presented as median (Me) and interquartile range [Q1; Q3]. Categorical variables are presented as *n* (%). The Mann–Whitney U test was used to compare the distributions of continuous variables and the two-sided Fisher’s exact test was used to compare categorical variables. For pairwise comparisons, when there were more than two samples, Holm’s correction for multiple comparisons was applied. The prevalence of dyslipidemia was standardized by sex and age, where 2010 All-Russian Population Census data were used as a reference. We considered outcomes statistically significant if the *p* value was <0.05.

## 3. Results

Figure 1 shows data on the prevalence of dyslipidemia and its types, depending on sex and age. In general, the prevalence of dyslipidemia (either hypercholesterolemia, hypertriglyceridemia, low HDL-C level, or any their combination) in the regions of the Russian Federation included into this study comprised 76.1% (75.5–76.7), with 76.9% (76.0–77.9) among men and 75.3% (74.6–76.0) among women (*p* < 0.001).

The most prevalent type of dyslipidemia was hypercholesterolemia (TC level ≥ 5.0 mmol/L), which, on average, was 58.2% and varied from 50 to 67% depending on the region [12]. Atherogenic shifts in the lipid profile as moderate hypercholesterolemia were mainly due to an increased (≥3.0 mmol/L) LDL-C level; its prevalence was quite high and amounted up to 61.8% among men and 58.9% among women. Hypertriglyceridemia (TG ≥ 1.7 mmol/L) was detected in every third man (30.1%) and in every fifth woman (21.4%). Low HDL-C prevalence amounted on average to 18.5%, varying from 1.97 to 29.3% in men and from 8.5 to 40.1% in women, depending on the region [12]. The frequency of detection of the apo B/apo AI ratio ≥1.0 varied in a very wide range: from 0–0.25 to 15.2%, averaging 5.3 and 3.8% in men and women, respectively. The prevalence of elevated levels of Lp(a) ≥ 30 mg/dL comprised 18.9% for the total cohort (18.0 and 19.7%, in men and women, respectively). A detailed analysis of the prevalence of dyslipidemia in various regions of the Russian Federation, differing in climate, geographic, economic, and ethnic characteristics, was presented elsewhere [12].

Figure 2A–F and Appendix A demonstrate the lipoprotein profile parameters in men and women, depending on age. Both men and women have a statistically significant increase in the TC level with age (Figure 2A); however, while women’s TC level increased gradually (from 4.72 ± 0.96 at the age of 25–34 years to 5.93 ± 1.2 mmol/L at 55–64 years), men reached the maximum TC level already at the age of 45–54 years. Interestingly, in each age decade, men and women differed in TC level; up to 45–54 years of age, the level of TC was higher in men; after that age, TC prevailed in women (*p* < 0.0001).

Variations in the TC level were largely due to the content of cholesterol transported within LDL (Figure 2B, Appendix A). At the same time, by the age of 45–54, the differences between sexes in LDL-C level disappeared, and in women of 55–64 years old, LDL-C as well as TC levels significantly exceeded those of men. A similar relationship for the level of apo B, the main protein of low-density lipoproteins, was found (Appendix A).

Both in men and women, a significant increase with age was also observed in the level of TG (Figure 2C, Appendix A). Women had a significantly lower TG level than men throughout the age range up to 55–64, when it slightly, but statistically significantly, exceeded the TG level in men (1.4 [1.04; 1.96] vs. 1.35 [0.98; 1.95] mmol/L, respectively; *p* = 0.013).

The level of HDL-C in women in the entire age range was significantly higher than in men (Figure 2D, Appendix A). No differences in HDL-C level between men of different ages were found; at the same time, a slight decrease in HDL-C level was observed in women by the age of 55–64 years, as compared to young ones. Similar relationships were obtained from the analysis of apo AI level—the major HDL protein (Appendix A). As a result, the apo B/apo AI ratio was significantly higher in men than in women in all age groups (Figure 2E, Appendix A), except at the age of 55–64 years, when a lower value of apo B/apo AI in young women (0.44 [0.35; 0.56]) increased up to 0.65 [0.5; 0.82] and became almost equal to the apo B/apo AI ratio in men (0.66 [0.49; 0.83]). As one can see from Figure 2F and Appendix A, both in men and women, the Lp(a) level increased with age; at the same time, while no differences between gender groups among young people (25–44 years old) have been observed, starting from 45 years old, women were characterized by a significantly higher Lp(a) level.

Table 1 shows the results of the analysis of lipoprotein profile parameters depending on the level of education. In all education groups, women had a higher level of TC than men. The highest level of TC was found in subjects with secondary education (mainly due to women: 5.57 ± 1.21 mmol/L), whereas the lowest level was detected in subjects (both sexes) with higher education. While the men with both primary and higher education had the same level of TC, in women, the differences between those with primary and higher education were significant. In men, the LDL-C level did not depend on the education level and was significantly lower than in women in groups with primary and secondary education; however, in subjects with higher education, it significantly exceeded that in women (3.36 ± 0.97 vs. 3.31 ± 1.02 mmol/L; *p* = 0.004).

There were no differences in TG level between men and women with primary education; in women, a higher level of education was associated with a reduced TG level. At the same time, men with secondary education were characterized by the highest TG level, whereas in women, it decreased with the level of education.

The highest level of Lp(a), regardless of sex, was found in subjects with primary education (higher in women than in men); at the same time, there were no differences in men differing by the level of education, whereas in women with higher education, the level of Lp(a) was significantly lower than in those with primary education.

Regardless of the education level, women had an expectedly higher level of HDL-C and HDL major protein apo AI. There were no differences in HDL-C level between men with different education levels, but serum apo AI concentration appeared to be highest in the group having secondary education. On the contrary, in women, both HDL-C level and apo AI concentrations were significantly higher in subjects with higher education compared to those who had primary education (1.50 ± 0.35 vs. 1.43 ± 0.34 mmol/L; *p* < 0.0001 for HDL-C, and 1.65 ± 0.39 vs. 1.60 ± 0.39 g/L; *p* = 0.032 for apo AI).

A gradual increase in apo AI and a decrease in apo B levels in women from primary to higher education were reflected in a decrease in apo B/apo AI ratio as the level of education increased. In contrast, men had the highest level of apo AI in persons with secondary education and a constant apo B across all groups, so the ratio of apo B/apo AI in men with secondary education turned out to be lowest (Table 1). Despite the statistically significant variations in the level of apo B obtained in this study between men and women with different levels of education, these fluctuations do not seem to be significant determinants of the association of atherogenic potential with the level of education.

Table 2 shows the profile of lipoproteins in the examined cohort, depending on the place of residence: urban or rural area. In men, the levels of TC and TG did not depend on whether they lived in urban or rural areas, whereas women living in rural areas had significantly higher levels of these parameters than urban residents. At the same time, the LDL-C level among men living in cities was significantly higher, while in women living in cities, it was significantly lower than among subjects living in rural areas. The same patterns were found in women for apo B level, whereas there were no differences in apo B between men, depending on the place of their residence. Regardless of sex, rural residents had higher levels of HDL-C and apo AI; as a result, the rural residents were characterized by a reduced apo B/apo AI ratio as compared to those living in urban areas. According to the level of Lp(a), men from urban and rural areas did not differ from each other, but women living in rural areas had a slightly higher level of Lp(a) than urban women did (13.30 [5.4; 27.6] vs. 11.30 [5; 28.3] mg/dL; *p* = 0.018).

Women from the low-income group had the highest levels of TC, LDL-C, TG, and apo B as compared with the high-income participants (Table 3). In the male cohort, no differences between subjects with different income levels in TC, TG, and apo B were found; the only exception was a slightly but statistically significantly increased level of LDL-C in the high-income group as compared with those with low income (3.40 ± 1.00 vs. 3.31 ± 1.01 mmol/L; *p* = 0.037). The results of the analysis of the relationship of Lp(a) level with the income showed its gradual decrease with the level of income for men, with differences only between low- and high-income subjects for women.

No differences in HDL-C level were detected between men with different income levels. On the contrary, the HDL-C level in women increased with their well-being. Apo AI showed an increase with income regardless of gender, and these changes were realized at lower values of apo B/apo AI ratio in high-income individuals regardless of gender (Table 3).

## 4. Discussion

This study was undertaken to compare the blood lipids levels in subjects of different sex and age in the representative population samples from 13 regions of the Russian Federation. It should be noted that these regions do not represent the Russian population as a whole; however, they have been selected to reflect different climates, geographical locations, levels of industrial development, and ethnic diversity, and were representative for each particular region.

Numerous papers have reported the data on lipid profile and dyslipidemia prevalence in different countries and regions worldwide, indicating the real need and importance of having data on the national characteristics of these disorders, including socioeconomic, behavioral, metabolic, and hereditary peculiarities [13,14,15,16,17,18]. However, in most studies, only trends of means of TC, HDL-C, and non-HDL-C have been assessed, and the analysis did not include other lipids of interest to researchers and clinicians. In contrast, in our study, we presented data not only on lipid levels, but also on major apolipoproteins, as well as on specific Lp(a), depending on sex, age, and three socioeconomic parameters (education, prosperity, and place of residence). It is very important because not only hypercholesterolemia due to high LDL-C but also the entire spectrum of lipid abnormalities, including low HDL-C, increased Lp(a) serum concentration, and hypertriglyceridemia, contribute to the level of risk [19].

The prevalence of elevated levels of TC and LDL-C levels and their mean levels in the regions of Russia turned out to be higher than in the USA [2,20]. A comparative study of lipid profiles in Russian and US men and women showed that, in Russia, the TC and LDL-C levels are worse, while TG and HDL-C are partly less atherogenic than in the USA [7].

There are accumulating data showing that the burden of ASCVD and dyslipidemia is increasing across developing countries: while age-specific ASCVD in many high-income countries is declining, the prevalence of plasma lipid disorders in the Asia-Pacific region is rising [14,17]; similar data were reported for Mexico [19]. In a recent review, Carrillo-Larco et al. demonstrated that the most common type of dyslipidemia in Latin America and the Caribbean region since 2005 is low HDL-C level (48%), followed by elevated TG (21%) and high LDL-C (20%) [21]. Xing et al. [22] reported a low prevalence of dyslipidemia in the northeast China region—35.8%, higher among urban residents than among rural ones (49.5 vs. 30.2%), and in women than in men (37.6 vs. 33.0%). The prevalence of elevated levels of TC (14.2%), TG (17.7%), LDL-C (5.7%), and low HDL-C (11.4%) [22] was much lower than that found in the ESSE-RF study. Of interest, the prevalence of high LDL-C and low HDL-C levels in China urban areas showed a 2.2-fold and 6.3-fold increase over the rural areas—values that significantly differed from those for the Russian population [12]. Gupta et al. (2017) reported that in India, a high TC level is present in 25–30% of urban and 15–20% of rural subjects [15]. This prevalence is lower than in high-income countries.

The multidirectional trends in the lipid profile parameters between industrially developed and developing countries have been revealed. In the United Arab Emirates, men had higher mean levels of TC and LDL-C than women did, which decreased with age in both groups. Similarly, men had higher mean TG levels than women did, which decreased sharply after 50 years of age but increased after the same age in women. The mean HDL-C level also varied depending on sex and age [16].

In the analysis of trends in serum TC levels for adults aged 25+ in 199 countries and regions from 1980 to 2008, it was demonstrated that in 2008, serum TC was highest in the high-income region including Australasia, North America, and Western Europe, whereas the lowest TC levels were detected in sub-Saharan Africa [9]. The analysis of trends from 1980 to 2018 in the mean level of TC, non-HDL-C, and HDL-C for 200 countries with more than 1000 population-based studies including more than 100 million participants aged 18+ [10,19] showed some increases in TC and non-HDL-C in low- and middle-income countries (especially in Asian countries), and decreases in high-income Western countries, including those from Europe, North America, and Australia.

Besides lipid parameters, in several regions included in the ESSE-RF study, we were able to measure the levels of major apolipoproteins apo AI and apo B and to calculate the apo B/apo AI ratio. In men, there was a uniform increase in the prevalence of a high apo B/apo AI ratio with increasing age, while in women, the prevalence did not change up to 44 years, with a sharp increase in people over 45 years; at the same time, if, in men, the prevalence of apo B/apo AI ≥ 1.0 increased by 2 times by 55 years, then in women at this age, the prevalence of a high apo B/apo AI ratio was already 4 times higher than in young people. There was no clear correlation between the level of education and the level of income for either men or women. At the same time, in both men and women, the frequency of increased apo B/apo AI ratio (≥1.0) in urban residents was 1.5–2 times higher than in rural ones [12]. Case-control studies have reported a significant association of ASCVD with elevated apo B levels and decreased apo A concentration [15].

Lp(a) is a strong cardiovascular risk predictor, and an elevation in its level (especially at levels exceeding 30–50 mg/dL) is a hereditary condition associated with increases in atherogenic, inflammation, and prothrombotic risk, for what should be considered as an independent ASCVD risk factor. Using ESSE-RF data, a significant association of Lp(a) serum level exceeding 9 mg/dL with the presence of myocardial infarction and coronary artery disease in adults aged 25–64 years has recently been shown [13]. A prospective observation is currently being performed within the follow-up step of the ESSE-RF. This will provide important information about Lp(a) as a prognostic factor for ASCVD in the Russian population.

Santo et al. (2019) showed that, in men, the higher the socioeconomic or educational status, the higher the TC, LDL-C, and TG levels and the lower the HDL-C levels, even after controlling for age, body mass index, arterial hypertension, smoking habits, and physical activity. In women, higher socioeconomic strata were associated with higher levels of TC and HDL-C, whereas lower levels of TC, LDL-C, and TG were found in individuals with higher education. In addition, individuals from the upper socioeconomic strata had higher levels of TC and LDL-C and were more than twice as likely to have multiple blood lipid changes [23].

Lara and Amigo [24] performed the cohort study and analyzed the effect of education on blood lipid profile in dependence on the income increase over the decade. A low level of education in Chile was associated with worse lipid profiles in women and better lipid profiles in men. In another study, it was reported that people with a low level of education were more likely to use less healthy eating and lifestyle patterns, and this may increase alongside the growth of their income without changes in the education level [25].

In our study, we did not obtain any pronounced associations between educational status and lipoprotein parameters in men, while women with higher education showed a significantly better lipid profile; the same trends were found when analyzing associations between lipids and prosperity. The influence of socioeconomic factors on lipid levels has been shown in epidemiological studies performed in developed American and European countries, also with different results in men and women: TC levels decreased in a gradient fashion with increasing socioeconomic and educational level; in women, these changes manifested to a greater extent. Thus, it is still not entirely clear whether there are differences in blood lipid levels in adults based on education status. Indeed, if, in Russia, there is a certain consistency between the level of education and income—the higher the education, the higher the well-being of people, in developing countries—there is some discrepancy between a low level of education and growing incomes allowing the purchase of unhealthy food [24]. Nonetheless, this information is necessary both for understanding the mechanisms underlying these disparities and for considering it for ASCVD prevention.

Taken together, our and the literature data indicate that every country or region has its own characteristics that determine the magnitude of epidemiological burden due to dyslipidemia and its atherosclerotic consequences. Wide variations in the levels of lipoproteins and in different types of dyslipidemia prevalence, which depend on sex, age, race/ethnicity, level of education and income, country, and place of residence, have been found. This means the need for specific information reflecting these features that could be obtained from population-based epidemiological studies, both cross-sectional and prospective ones. Thus, strategies aimed at correcting dyslipidemia should be developed by taking into account sex and age variations, as well as socioeconomic and regional differences.

As a strength of our study, the possibility of measurement and analysis of the protein compounds of lipoproteins in several regions included in the ESSE-RF study should be pointed out, which sets our study apart from almost from all abovementioned.

Among the limitations of this study, we should indicate that the sample from 13 regions was not representative for the male and female population of the whole country, but for each region participating in the ESSE-RF study. In addition, our cohort did not include seriously ill patients, because data collection at home was not provided. In addition, persons leading an antisocial lifestyle were not included.

## 5. Conclusions

To conclude, in this paper based on the ESSE-RF study, we analyzed and found some peculiarities on lipids and lipoprotein profiles and the prevalence of atherogenic dyslipidemias in the general population from several Russian regions differing in climate and geography, taking into account sex, age, education and income level, as well as settlement (urban or rural). These data indicate the need to develop ASCVD preventive strategies based on our own data.

## Figures and Tables

**Figure 1 ijerph-19-00931-f001:**
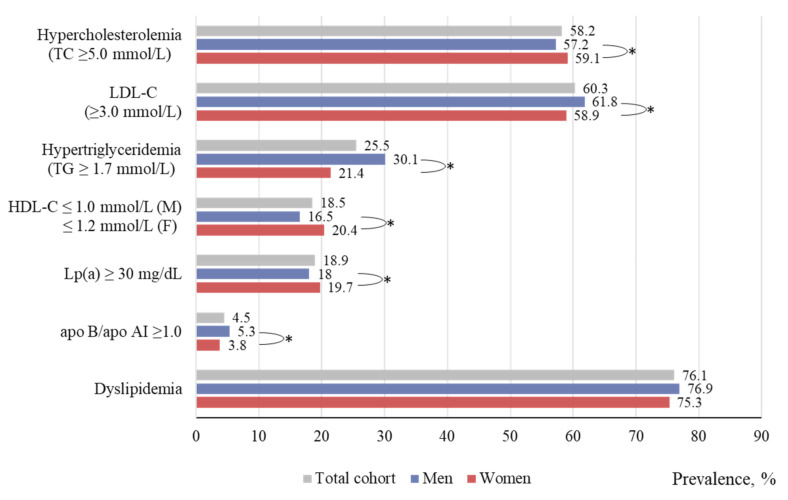
Dyslipidemia prevalence according to ESSE-RF study data. (*) indicates the significant difference between men and women in the corresponding age group.

**Figure 2 ijerph-19-00931-f002:**
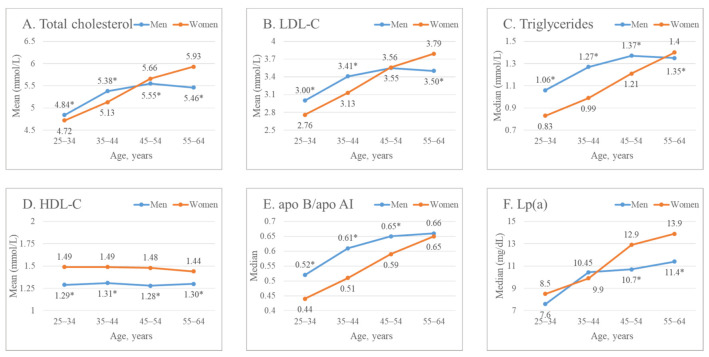
Sex- and age-specific levels of: (**A**) total cholesterol; (**B**) LDL-C; (**C**) Triglycerides; (**D**) HDL-C; (**E**) apoB/ apo AI; (**F**) Lp(a). (*) indicates the significant difference between men and women in the corresponding age group.

**Table 1 ijerph-19-00931-t001:** Lipoprotein profile parameters in men and women differing in education level among populations from regions of the Russian Federation: ESSE-RF study.

Level of Education	Total Cohort	Men	Women	*p*
	Total cholesterol (M ± SD), mmol/L			
Primary	5.39 ± 1.21 *n* = 919	5.22 ± 1.16 *n* = 390	5.52 ± 1.22 ^c^ *n* = 529	0.000
Secondary	5.48 ± 1.20 ^c^ *n* = 11078	5.32 ± 1.17 *n* = 4033	5.57 ± 1.21 ^c^ *n* = 7045	0.000
Higher	5.34 ± 1.15 ^b^ *n* = 9168	5.29 ± 1.15 *n* = 3515	5.37 ± 1.15 ^a,b^ *n* = 5653	0.002
	LDL-cholesterol (M ± SD), mmol/L			
Primary	3.45 ± 1.04 ^c^ *n* = 919	3.35 ± 1.01 *n* = 390	3.52 ± 1.05 ^c^ *n* = 529	0.004
Secondary	3.45 ± 1.04 ^c^ *n* = 11079	3.37 ± 1.01 *n* = 4034	3.49 ± 1.05 ^c^ *n* = 7045	0.000
Higher	3.33 ± 1.00 ^a,b^ *n* = 9168	3.36 ± 0.97 *n* = 3515	3.31 ± 1.02 ^a,b^ *n* = 5653	0.004
	Triglycerides (Me [Q1; Q3]), mmol/L			
Primary	1.26 [0.89; 1.82] ^c^ *n* = 919	1.21 [0.86; 1.82] *n* = 390	1.30 [0.93; 1.80] ^c^ *n* = 529	0.111
Secondary	1.26 [0.89; 1.82] ^c^ *n* = 11078	1.29 [0.90; 1.91] *n* = 4034	1.24 [0.89; 1.77] ^c^ *n* = 7044	0.000
Higher	1.12 [0.79; 1.64] ^a,b^ *n* = 9168	1.23 [0.88; 1.85] *n* = 3515	1.04 [0.75; 1.52] ^a,b^ *n* = 5653	0.000
	HDL-cholesterol (M ± SD), mmol/L			
Primary	1.37 ± 0.35 ^c^ *n* = 919	1.30 ± 0.34 ^b^ *n* = 390	1.43 ± 0.34 ^c^ *n* = 529	0.000
Secondary	1.40 ± 0.35 ^c^ *n* = 11079	1.30 ± 0.33 *n* = 4034	1.45 ± 0.35 ^c^ *n* = 7045	0.000
Higher	1.42 ± 0.35 ^a,b^ *n* = 9168	1.29 ± 0.33 *n* = 3515	1.50 ± 0.35 ^a,b^ *n* = 5653	0.000
	Lp(a) (Me [Q1; Q3]), mg/dL			
Primary	13.00 [5.50; 28.25] ^c^ *n* = 555	11.60 [4.27; 26.52] *n* = 224	13.70 [6.70; 29.05] ^c^ *n* = 331	0.011
Secondary	11.30 [5.00; 27.28] ^c^ *n* = 5338	9.70 [4.20; 23.40] *n* = 1771	12.40 [5.40; 29.00] ^c^ *n* = 3567	0.000
Higher	10.60 [4.50; 25.40] ^a,b^ *n* = 4537	9.80 [4.20; 22.10] *n* = 1768	10.90 [4.70; 27.20] ^a,b^ *n* = 2769	0.002
	apo AI (M ± SD), g/L			
Primary	1.55 ± 0.38 ^b^ *n* = 587	1.47 ± 0.35 ^b^ *n* = 239	1.60 ± 0.39 *n* = 348	0.000
Secondary	1.60 ± 0.42 ^a^ *n* = 5846	1.55 ± 0.49 ^a,c^ *n* = 1937	1.62 ± 0.38 ^c^ *n* = 3909	0.000
Higher	1.59 ± 0.39 *n* = 5374	1.49 ± 0.37 ^b^ *n* = 2081	1.65 ± 0.39 ^b^ *n* = 3293	0.000
	apo B (M ± SD), g/L			
Primary	0.95 ± 0.27 ^c^ *n* = 588	0.93 ± 0.25 *n* = 239	0.96 ± 0.27 ^c^ *n* = 349	0.18
Secondary	0.94 ± 0.26 ^c^ *n* = 5870	0.92 ± 0.26 ^c^ *n* = 1948	0.95 ± 0.27 ^c^ *n* = 3922	0.001
Higher	0.89 ± 0.25 ^a,b^ *n* = 5393	0.90 ± 0.24 ^b^ *n* = 2088	0.89 ± 0.26 ^a,b^ *n* = 3305	0.002
	apo B/apo AI (Me [Q1; Q3])			
Primary	0.62 [0.47; 0.78] ^b,c^ *n* = 587	0.63 [0.50; 0.80] *n* = 239	0.62 [0.46; 0.77] ^c^ *n* = 348	0.161
Secondary	0.59 [0.45; 0.76] ^a,c^ *n* = 5846	0.59 [0.45; 0.78] *n* = 1937	0.58 [0.45; 0.75] ^c^ *n* = 3909	0.023
Higher	0.56 [0.43; 0.73] ^a,b^ *n* = 5373	0.61 [0.46; 0.78] *n* = 2080	0.53 [0.40; 0.69] ^a,b^ *n* = 3293	0.000

^a,b,c^—*p* < 0.05, where the group with primary education is indicated by ^a^, with secondary—by ^b^, and with higher education—by ^c^; *p*—significance of differences between men and women.

**Table 2 ijerph-19-00931-t002:** Lipoprotein profile parameters in men and women differing in place of residence among populations from regions of the Russian Federation: ESSE-RF study.

Place of Residence	Total Cohort	Men	Women	*p*
	Total cholesterol (M ± SD), mmol/L			
Urban	5.40 ± 1.18 ^b^ *n* = 17133	5.30 ± 1.15 *n* = 6479	5.45 ± 1.19 ^b^ *n* = 10,654	0.000
Rural	5.50 ± 1.20 ^a^ *n* = 4032	5.31 ± 1.19 *n* = 1459	5.61 ± 1.19 ^a^ *n* = 2573	0.000
	LDL-cholesterol (M ± SD), mmol/L			
Urban	3.38 ± 1.02 ^b^ *n* = 17133	3.37 ± 0.99 ^b^ *n* = 6479	3.39 ± 1.04 ^b^ *n* = 10,654	0.815
Rural	3.45 ± 1.03 ^a^ *n* = 4032	3.31 ± 1.02 ^a^ *n* = 1459	3.53 ± 1.03 ^a^ *n* = 2573	0.000
	Triglycerides (Me [Q1; Q3]), mmol/L			
Urban	1.19 [0.84; 1.73] ^b^ *n* = 17,132	1.26 [0.89; 1.87] *n* = 6479	1.14 [0.81; 1.65] ^b^ *n* = 10,653	0.000
Rural	1.23 [0.88; 1.78] ^a^ *n* = 4032	1.26 [0.89; 1.93] *n* = 1459	1.21 [0.87; 1.72] ^a^ *n* = 2573	0.003
	HDL-cholesterol (M ± SD), mmol/L			
Urban	1.40 ± 0.35 ^b^ *n* = 17,133	1.29 ± 0.33 ^b^ *n* = 6479	1.47 ± 0.35 ^b^ *n* = 10,654	0.000
Rural	1.43 ± 0.34 ^a^ *n* = 4032	1.32 ± 0.32 ^a^ *n* = 1459	1.49 ± 0.34 ^a^ *n* = 2573	0.000
	Lp(a) (Me [Q1; Q3]), mg/dL			
Urban	10.70 [4.70; 26.70] ^b^ *n* = 7914	9.70 [4.10; 22.80] *n* = 2997	11.30 [5.00; 28.30] ^b^ *n* = 4917	0.000
Rural	12.50 [5.10; 26.83] ^a^ *n* = 2516	10.40 [4.40; 23.95] *n* = 766	13.30 [5.40; 27.60] ^a^ *n* = 1750	0.000
	apo AI (M ± SD), g/L			
Urban	1.57 ± 0.40 ^b^ *n* = 9290	1.50 ± 0.43 ^b^ *n* = 3490	1.61 ± 0.37 ^b^ *n* = 5800	0.000
Rural	1.67 ± 0.41 ^a^ *n* = 2516	1.59 ± 0.37 ^a^ *n* = 766	1.70 ± 0.42 ^a^ *n* = 1750	0.000
	apo B (M ± SD), g/L			
Urban	0.92 ± 0.27 ^b^ *n* = 9334	0.91 ± 0.26 *n* = 3508	0.93 ± 0.27 ^b^ *n* = 5826	0.081
Rural	0.90 ± 0.23 ^a^ *n* = 2516	0.90 ± 0.23 *n* = 766	0.89 ± 0.23 ^a^ *n* = 1750	0.323
	apo B/apo AI (Me [Q1; Q3])			
Urban	0.59 [0.45; 0.76] ^b^ *n* = 9289	0.62 [0.47; 0.8] ^b^ *n* = 3489	0.58 [0.44; 0.74] ^b^ *n* = 5800	0.000
Rural	0.53 [0.41; 0.68] ^a^ *n* = 2516	0.55 [0.44; 0.73] ^a^ *n* = 766	0.52 [0.41; 0.67] ^a^ *n* = 1750	0.000

^a,b^—*p* < 0.05, where the group of participants from an urban area is indicated by ^a^ and the group from a rural area is indicated by ^b^; *p*—significance of differences between men and women.

**Table 3 ijerph-19-00931-t003:** Lipoprotein profile parameters in men and women differing in income level among populations from regions of the Russian Federation: ESSE-RF study.

Income Level	Total Cohort	Men	Women	*p*
	Total cholesterol (M ± SD), mmol/L			
Low	5.51 ± 1.21 ^b,c^ *n* = 3539	5.28 ± 1.18 *n* = 941	5.60 ± 1.21 ^b,c^ *n* = 2598	0.000
Middle	5.42 ± 1.18 ^a,c^ *n* = 13,547	5.29 ± 1.15 *n* = 4938	5.49 ± 1.19 ^a,c^ *n* = 8609	0.000
High	5.32 ± 1.14 ^a,b^ *n* = 3784	5.33 ± 1.15 *n* = 1907	5.31 ± 1.13 ^a,b^ *n* = 1877	0.585
	LDL-cholesterol (M ± SD), mmol/L			
Low	3.46 ± 1.04 ^b,c^ *n* = 3539	3.31 ± 1.01 *n* = 941	3.52 ± 1.04 ^b,c^ *n* = 2598	0.000
Middle	3.40 ± 1.02 ^a,c^ *n* = 13548	3.35 ± 0.98 *n* = 4939	3.42 ± 1.04 ^a,c^ *n* = 8609	0.004
High	3.33 ± 1.01 ^a,b^ *n* = 3784	3.40 ± 1.00 *n* = 1907	3.26 ± 1.01 ^a,b^ *n* = 1877	0.000
	Triglycerides (Me [Q1; Q3]), mmol/L			
Low	1.28 [0.91; 1.84] ^b,c^ *n* = 3539	1.27 [0.92; 1.9] ^b,c^ *n* = 941	1.28 [0.91; 1.81] ^b,c^ *n* = 2598	0.631
Middle	1.20 [0.85; 1.75] ^a,c^ *n* = 13,548	1.26 [0.89; 1.89] *n* = 4939	1.16 [0.83; 1.68] ^a,c^ *n* = 8609	0.000
High	1.12 [0.78; 1.63] ^a,b^ *n* = 3783	1.26 [0.87; 1.87] *n* = 1907	0.99 [0.73; 1.43] ^a,b^ *n* = 1876	0.000
	HDL-cholesterol (M ± SD), mmol/L			
Low	1.39 ± 0.34 ^b,c^ *n* = 3539	1.29 ± 0.34 ^b,c^ *n* = 941	1.42 ± 0.34 ^b,c^ *n* = 2598	0.000
Middle	1.41 ± 0.35 ^a^ *n* = 13,548	1.30 ± 0.33 ^a,c^ *n* = 4939	1.47 ± 0.35 ^a,c^ *n* = 8609	0.000
High	1.41 ± 0.35 ^a^ *n* = 3784	1.29 ± 0.32 ^a,b^ *n* = 1907	1.53 ± 0.34 ^a,b^ *n* = 1877	0.000
	Lp(a) (Me [Q1; Q3]), mg/dL			
Low	12.00 [5.30; 29.75] ^b,c^ *n* = 1794	12.40 [5.30; 32.65] ^b,c^ *n* = 439	11.90 [5.30; 28.40] ^c^ *n* = 1355	0.899
Middle	11.30 [4.90; 27] ^a,c^ *n* = 6475	9.60 [4.10; 22.40] ^a^ *n* = 2185	12.20 [5.30; 28.40] ^c^ *n* = 4290	0.000
High	9.50 [4.30; 22.02] ^a,b^ *n* = 1952	9.50 [4.12; 20.35] ^a^ *n* = 1030	9.60 [4.50; 25.00] ^a,b^ *n* = 922	0.113
	apo AI (M ± SD), g/L			
Low	1.51 ± 0.37 ^b,c^ *n* = 1952	1.45 ± 0.39 ^b,c^ *n* = 469	1.54 ± 0.36 ^b,c^ *n* = 1483	0.000
Middle	1.60 ± 0.39 ^a^ *n* = 7244	1.53 ± 0.39 ^a^ *n* = 2431	1.64 ± 0.38 ^a,c^ *n* = 4813	0.000
High	1.59 ± 0.40 ^a^ *n* = 2372	1.50 ± 0.36 ^a^ *n* = 1230	1.70 ± 0.41 ^a,b^ *n* = 1142	0.000
	apo B (M ± SD), g/L			
Low	0.93 ± 0.25 ^b,c^ *n* = 1962	0.92 ± 0.25 *n* = 475	0.94 ± 0.25 ^c^ *n* = 1487	0.268
Middle	0.92 ± 0.26 ^a,c^ *n* = 7272	0.91 ± 0.25 *n* = 2440	0.93 ± 0.27 ^c^ *n* = 4832	0.012
High	0.89 ± 0.25 ^a,b^ *n* = 2378	0.91 ± 0.25 *n* = 1233	0.86 ± 0.24 ^a,b^ *n* = 1145	0.000
	apo B/apo AI (Me [Q1;Q3])			
Low	0.63 [0.49; 0.79] ^b,c^ *n* = 1952	0.65 [0.50; 0.82] ^b,c^ *n* = 469	0.62 [0.48; 0.78] ^b,c^ *n* = 1483	0.010
Middle	0.57 [0.44; 0.74] ^a,c^ *n* = 7243	0.60 [0.45; 0.78] ^a^ *n* = 2430	0.56 [0.43; 0.73] ^a,c^ *n* = 4813	0.000
High	0.55 [0.42; 0.73] ^a,b^ *n* = 2372	0.61 [0.47; 0.79] ^a^ *n* = 1230	0.50 [0.39; 0.65] ^a,b^ *n* = 1142	0.000

^a,b,c^—*p* < 0.05, where the group with a low income level is indicated by ^a^, with middle income level—by ^b^, and with high income level—by ^c^; *p*—significance of differences between men and women.

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
