# Peer review of "Lipoprotein Profile in Populations from Regions of the Russian Federation: ESSE-RF Study"

_ijerph, 2022, doi:10.3390/ijerph19020931_

Round 1
Reviewer 1 Report
The manuscript ijerph-1503226 is interesting and takes into account the levels of several circulating lipid profile parameters that can be used for risk assessment associated with cardiovascular disorders. This topic is particularly interesting also for the purpose of preventive medicine. Although there are similar works in the literature, this manuscript takes into consideration 13 states of the Russian federation that are heterogeneous in terms of ethnicity, climate and habits and the parameters analyzed for age, sex, level of education and financial income.
Obviously the work takes into consideration a group of healthy population having to answer also to the questionnaires. This fact should be expanded in the section on employment restrictions.
Compared to other published works, the evaluation of lipoproteins and apolipoproteins allows to have a more complete picture of the lipid status of the cohort.
Minor comments:
a) The analysis of the different parameters by age, reveals a similar trend for all or in women there is a continuous increase with age while for men there is a maximum for the 45-54 group and then remains constant.
Authors should comment extensively on this phenomenon.
b) Could the authors do an analysis for eating habits and / or climate? These factors could certainly give further surprises and indications.
Author Response
Dear Reviewer,
Thank you very much for reading carefully and for your valuable comments which we've tried to incorporate into the revised version.
Attached, plese, find our reply.
Sincerely yours,
Victoria A Metelskaya, on behalf of the manuscripts authors

Reviewer 2 Report
Victoria Metelskaya and colleagues' manuscript described the lipid profile of the Russian population, taking into consideration sex and education. They found that demographic variables are helpful in asses atherosclerotic risk.
Bellow, you can find comments to improve the manuscript.
- I noticed that your group recently published the following manuscripts:
"Lipoprotein(a) in an adult sample from the Russian population: distribution and association with atherosclerotic cardiovascular diseases." DOI: https://doi.org/10.5114/aoms/131089.
"The Prevalence of Heterozygous Familial Hypercholesterolemia in Selected Regions of the Russian Federation: The FH-ESSE-RF Study." doi: 10.3390/jpm11060464.
Is the population analyzed in both articles the same one of this manuscript? If yes, can you clarify if you are performing the secondary analysis in this manuscript?
- In the Material and Methods section, it would be good to add more details about study design.
- At line 97, add how the centrifugation was performed (example: where the blood was collected; how many rpm or rcf; how many minutes the samples were centrifuged).
- Add demographic table of the population on the results.
- Lines 228 to 234 and at the Discussion topic, the space between lines is not the same as other parts of the manuscript.
- The conclusion must be improved because it is not congruent with your aims.
Author Response
Dear Reviewer,
Thank you very much for reading our manuscript carefully and for your valuable comments which we've tried to incorporate into the revised version
Attached, please, find our reply
Cordially,
Victoria A Metelskaya, on behalf of manuscript authors

Round 2
Reviewer 2 Report
Dear Victoria A Metelskaya,
Thank you for your answers and for addressing the comments.
There are some minor corrections to be made at line 83 (population), line 85 (regions), line 86 (polyclinic), line 87 (secondary), line 89 (sampling).
Please, review the whole manuscript for typos before re-submission.
Author Response
Dear Reviewer,
Thank you very much again for careful reading of our manuscript, and for all useful comments
All corrections are done, uncluding check-up of typos
On behalf of all authors,
Cordially,
Victoria Metelskaya